A highly pathogenic porcine reproductive and respiratory syndrome virus candidate vaccine based on Japanese encephalitis virus replicon system

Hu Pingsheng 1
Chen Xiaoming 1
Huang Lihong 1
Liu Shukai 1
Zang Fuyu 1
Xing Jinchao 1
Zhang Youyue 1
Liang Jiaqi 1
Zhang Guihong 1
Liao Ming mliao@scau.edu.cn 1 2 3
Qi Wenbao qiwenbao@scau.edu.cn 1 2 3
1 National and Regional Joint Engineering Laboratory for Medicament of Zoonoses Prevention and Control, College of Veterinary Medicine, South China Agricultural University , Guangzhou , China
2 Key Laboratory of Zoonoses, Key Laboratory of Animal Vaccine Development, Ministry of Agriculture , Guangzhou , China
3 Key Laboratory of Zoonoses Prevention and Control of Guangdong Province, Ministry of Agriculture , Guangzhou , China
Flores-Valdez Mario Alberto
Electronic publication date: 2017 Jul 20
Publication date: 2017
Volume: 5
Electronic Location ID: e3514
Received 2017 May 6; Accepted 2017 Jun 7
Copyright: ©2017 Hu et al.
Copyright year: 2017
Copyright holder: Hu et al.
License: This is an open access article distributed under the terms of the Creative Commons Attribution License, which permits unrestricted use, distribution, reproduction and adaptation in any medium and for any purpose provided that it is properly attributed. For attribution, the original author(s), title, publication source (PeerJ) and either DOI or URL of the article must be cited.
License URL: https://creativecommons.org/licenses/by/4.0/

Keywords: JEV replicon, HP-PRRSV, Vaccine

Funding: National Natural Science Foundation of China 31272563 30800827 National Key Research and Development Program of China 2016YFD0500405 Special Support Plan of Guangdong Province in science and technology for talents This work was supported by the National Natural Science Foundation of China (31272563, 30800827), and the National Key Research and Development Program of China (2016YFD0500405). WQ and ML are supported by the “Special Support Plan of Guangdong Province in science and technology for talents”. There was no additional external funding received for this study. The funders had no role in study design, data collection and analysis, decision to publish, or preparation of the manuscript.

==============================
In the swine industry, porcine reproductive and respiratory syndrome (PRRS) is a highly contagious disease which causes heavy economic losses worldwide. Effective prevention and disease control is an important issue. In this study, we described the construction of a Japanese encephalitis virus (JEV) DNA-based replicon with a cytomegalovirus (CMV) promoter based on the genome of Japanese encephalitis live vaccine virus SA14-14-2, which is capable of offering a potentially novel way to develop and produce vaccines against a major pathogen of global health. This JEV DNA-based replicon contains a large deletion in the structural genes (C-prM-E). A PRRSV GP5/M was inserted into the deletion position of JEV DNA-based replicons to develop a chimeric replicon vaccine candidate for PRRSV. The results showed that BALB/c mice models with the replicon vaccines pJEV-REP-G-2A-M-IRES and pJEV-REP-G-2A-M stimulated antibody responses and induced a cellular immune response. Analysis of ELSA data showed that vaccination with the replicon vaccine expressing GP5/M induced a better antibodies response than traditional DNA vaccines. Therefore, the results suggested that this ectopic expression system based on JEV DNA-based replicons may represent a useful molecular platform for various biological applications, and the JEV DNA-based replicons expressing GP5/M can be further developed into a novel, safe vaccine candidate for PRRS.

Introduction

Porcine reproductive and respiratory syndrome (PRRS) is a disease that causes reproductive failures in sows and respiratory syndromes in pigs of all ages (Done & Paton, 1995). In China, the first emergence of PRRS occurred in 1995, and the causative agent PRRSV was first isolated in 1996. In 2006, a new strain of PRRSV appeared in many provinces, named highly pathogenic PRRSV (HP-PRRSV). Since then, it has been circulating and predominating in the field (Li et al., 2007; Tong et al., 2007). The GP5 protein of PRRSV (encoded by ORF5) is one of the major antigens expressed on virion surfaces and, along with matrix (M) protein, are thought to be the most important targets of protective antibodies (Li & Murtaugh, 2012; Wang et al., 2007). Vaccination has been an effective way to control PRRS; and there are two main types of PRRS vaccines since they were first reported, modified live-attenuated vaccines (MLVs) and inactivated virus vaccines (Charerntantanakul, 2012). Three commercial HP-PRRS modified live virus vaccines (HP PRRS MLVs), TJM-F92, HuN4-F112, and JXA-1R, were introduced into the Chinese swine industry, all providing good protection against HP-PRRSV infection (Leng et al., 2012; Tian et al., 2009; Yu et al., 2013). However, both kinds of the vaccine, MLVs and inactivated virus vaccines, have inherent drawbacks. Although MLVs perform well against homologous infection (Jeong et al., 2016; Linhares et al., 2012), MLVs show poor protection to heterologous strains and cannot prevent PRRSV to disseminate through the placenta (Kimman et al., 2009). Furthermore, China and Denmark each have related reports that MLVs might revert to the high virulence strain (Botner et al., 1997; Jiang et al., 2015; Opriessnig et al., 2002). Inactivated virus vaccines are safer than MLVs, but the inactivated virus vaccines provide limited protective immunity against PRRSV infection (Renukaradhya et al., 2015; Scortti et al., 2007). Moreover, more time and much more work are needed to develop both kinds of vaccine. There is an urgent need to develop vaccines against PRRS that are more effective, safe, and can be more quickly obtained, especially when new phenotypes of the PRRSV emerge.

Virus replicons hold tremendous promise as vaccine candidates because they can replicate autonomously and effectively express foreign proteins without being infectious (Kato & Hishiki, 2016). Because replicons lack the viral structural proteins genome, non-infectious replicons are as safe as a conventional inactive vaccine. At the same time, they can elicit a robust and broad reactive immune response (Aberle et al., 2005). In recent years, many flaviviruses virus replicons have been used as the expressing system for foreign genes or have been developed into vaccine candidates (Cao et al., 2011; Harvey et al., 2003). Strikingly, research has shown that the use of replicons as an expression system to create vaccines can support dual protection, both to itself and the exogenous virus (Huang et al., 2015; Yang et al., 2012).

Here, we constructed JEV DNA-based replicons with autonomous replication and effective expression of foreign proteins, which failed to generate infectious virus progeny in non-complementing cells. Moreover, we expressed GP5/M proteins of HP-PRRSV on these JEV DNA-based replicons, which elicited a humoral and cellular immune response to PRRSV in a BALB/c mice model. Thus, the results suggest that the JEV replicons are an efficient expression system for foreign proteins, and the chimeric PRRSV vaccine based on JEV replicons could be developed as a potential vaccine candidate against HP-PRRS and JEV.

Material and Methods

Cell lines, virus strain, vectors, and antibody

BHK-21 cells, a baby hamster kidney cell line, and 293T cells, a human embryonic kidney cells line, were obtained from the Key Laboratory of Animal Disease Control and Prevention, the Ministry of Agriculture, China. Cells were grown in Dulbecco’s Modified Eagle Medium (DMEM; Gibco, Invitrogen, Waltham, MA, USA) supplemented with 10% fetal bovine serum (FBS; Gibco, Billings, MT, USA). The JEV vaccine strain SA-14-14-2 was also provided by the Key Laboratory of Animal Disease Control and Prevention, the Ministry of Agriculture, China. The HP-PRRSV strain, XH-GD, is stored at the Key Laboratory of Animal Disease Control and Prevention of the Ministry of Agriculture, China. It was isolated in the Guangdong province in 2007 and is one of the epidemic HP-PRRSV strains in Guangdong, exhibiting 99.1% nucleotide sequence identity to strain JXA1. Plasmid pEGFP-N1 was purchased from Biosciences Clontech. Plasmids pIRES1-neo and pCAGGS were kindly provided by ZhiGao Bu (State Key Laboratory of Veterinary Biotechnology, Harbin Veterinary Research Institute, Chinese Academy of Agricultural Sciences, Harbin, China).

The monoclonal antibodies against the GP5 protein of PRRSV were kindly provided by Researcher GuangZhi Tong (Shanghai Veterinary Research Institute, Chinese Academy of Agricultural Sciences, Shanghai, China). Mouse polyclonal antibody against the NS1 protein of JEV was prepared and provided by the Key Laboratory of Animal Disease Control and Prevention of the Ministry of Agriculture. IRDye® 800-labeled anti-mouse IgG secondary antibodies were obtained from Rockland. Fluorescein isothiocyanate (FITC)-conjugated goat anti-mouse IgG were obtained from SIGMA.

Plasmid constructs

The JEV replicons plasmid vector is depicted in Fig. 1. All the primers used to construct the recombinant plasmid are shown in Tables 1 and 2. Fragments 5′UTR-c23,B,C, and II were amplified from the cNDA of JEV genome SA14-14-2 using primers in Table 1. The CMV promoter and fragment D were amplified from plasmid pEGFP-N1 using primers in Table 1. Fragment A was amplified from the CMV promoter and 5′UTR-c23 by overlap PCR using primers pCMV-F and pJEV164-olR. Then, fragments A and B were used as template to produce fragment I through overlap PCR using primers pCMV-F and pJEV5706R. Amplified fragment III using primers pJEV9117F and HDVr-pA-R in the same way as shown in Fig. 1. Finally, DNA fragment I, fragment II, and fragment III were subcloned into the Low copy plasmid named pJEV-REP. Fragment EGFP, both without and with the termination codon, were amplified from plasmid pEGFP-N1 using primers EGFP-F and EGFP-R or EGFP-F and EGFP-R-taa. Fragment IRES was amplified from the plasmid pIRES1-neo using primers IRES-F and IRS-588R. EGFP-IRES was amplified from the fragment IRES and fragment EGFP that includes the termination codon through overlap PCR using primers EGFP-F and IRES-588R. Finally, fragment EGFP without the termination codon and EGFP-IRES were digested by SalI and SpeI and then cloned into pJEV-REP namely pJEV-REP-GFP and pJEV-REP-GFP-IRES, respectively.

Figure 1 Schematic drawing of replicons.

(A) Strategy to construct the JEV replicon. An in-frame deletion in the structural region of C-prM-E was from nt 165 to 2,402 and SpeI and SalI site was generated at the junction of the deletion position. (B and C) Schematic drawing of replicons contain report gene. (B) The EGFP fragment without a termination code and initiation code was engineered into the deletion position at SpeI and SalI. (C) The IRES-EGFP fragment, containing a termination code at the end of EGFP and an initiation code at the front of E25, which was engineered into the deletion position at the SpeI and SalI site.

Table 1 Primers used for construction of the subgenomic replicons of JEV.

Primer	Sequences(5′–3′)	
pCMV-F	TTTTTGGCGGCCGCTAGTTATTAATAGTAATCAATTACGG	
pCMV-R	GTTCACACAGATAAACTTCTGGATCTGACGGTTCACTAAACCAGCTCTG	
5′UTR-c23		
pJEV1-F	AGAAGTTTATCTGTGTGAACTTCTTGG	
pJEV164-olR	CTCGGTCGACGGTGGTAACACTAGTGCGGGGTAGGCCGCGTTTCAGC	
Fragment B		
PJEV2403-olF	ACTAGTGTTACCACCGTCGACCGAGACCGATCAATTGCTTTGG	
pJEV5706R	TTACGCTCGCCACAAACCAC	
Fragment II		
pJEV-5557F	CGACCCCGCCTGGAACCACG	
pJEV-9155R	GAACCCCAAAGCTTCAAACTCTAGA	
Fragment C		
pJEV9117F	CTTGGAGCACGGTATCTAGAGTTTG	
pJEV10976R	AGCCGGCGCCAGCGAGGAGGCTGGGACCATGCCGGCCATCAGGAGATCCTGTGTTCTTCCTCACCAC	
Fragment D		
HDVr-pA-F	CCTGATGGCCGGCATGGTCCCAGCCTCCTCGCTGGCGCCGGCTAACTTGTTTATTGCAGCTTA	
HDVr-pA-R	CGAGGTACCTTGTCCAAACTCATCAATGTATCTTA	
Notes.

Lined part represent for restriction enzymes.

Table 2 Primers used for construction of the subgenomic replicons of JEV.

Primers used for construction of different plasmid and primers used for plus or minus detection of replicon RNA.

Primer	Sequences(5′–3′)	
EGFP-F	CCGCACTAGTATGGTGAGCAAGGGCGAGG	
EGFP-R	CTCGGTCGACCTTGTACAGCTCGTCCATGC	
EGFP-R-taa	GGGGGAGGGAGAGGGGCGTTACTTGTACAGCTCGTCCATGCC	
IRES-F	CGCCCCTCTCCCTCCCCC	
IRES-588R	CTCGGTCGACCATGTTGTGGCAAGCTTATCATCGTGTT	
2403F	CGAGACCGATCAATTGCTTTGG	
2609R	CTTCGCTAGGGATCTGGGCGTTTCTGG	
GP5F	CCGCACTAGTATGTTGGGGAAGTGCTTGACCGCGT	
GP5R-FMDV2A-R	CTCAACGTCTCCCGCCAACTTGAGGAGGTCGAAGTTCAGAAGCTGGAGACGACCCCATTGTTCTGCT	
MF-FMDV2A-F	GACCTCCTCAAGTTGGCGGGAGACGTTGAGTCCAACCCTGGGCCTATGGGGTCGTCTCTAGACGA	
MR-SalI	CCTTTGTCGACTTTGGCATATTTAACAAGGTTTACCACT	
M-R	GGGGGAGGGAGAGGGGCGTTATTTGGCATATTTAACAAGGTTTACCACT	
GP5F-EcoRI	CCTTTGAATTCATGTTGGGGAAGTGCTTGACCGCGT	
MR-XhoI	CCTTTCTCGAGTTATTTGGCATATTTAACAAGGTTTACCACT	
IRES-F-SpeI-TAA	CCGCACTAGTTAACGCCCCTCTCCCTCCCCC	
Notes.

Lined part represent for restriction enzymes.

Fragments GP5 (primers GP5F and GP5R-FMDV2A-R) and M (primers MF-FMDV2A-F and MR-SalI) were amplified from the PRRSV genome. Fragment G-2A-M was amplified from fragments GP5 and M through overlap PCR using primers GP5F and MR-SalI. Then, G-2A-M as a template to amplify G-2A-MR using primers GP5F and M-R. Fragments G-2A-MR and IRES were used as templates to produce G-2A-M-IRES through overlap PCR using primer GP5F and IRES-588R. Finally, G-2A-M-IRES and G-2A-M were digested by SalI and SpeI and cloned into pJEV-REP vector, named pJEV-REP-G-2A-M-IRES and pJEV-REP-G-2A-M, respectively. Using IRES-F-SpeI-TAA and IRES-588R as primers, we amplified IRES-SS from pJEV-REP-G-2A-M-IRES. They were then digested by SalI-HF and SpeI and cloned into a pJEV-REP vector named pJEV-REP-IRES. Fragment G-2A-M-EX was amplified from pJEV-REP-G-2A-M using primers GP5F-EcoRI and MR-XhoI. Then, fragment G-2A-M-EX was digested by EcoRI and XhoI and cloned into a eukaryotic expression vector pCAGGS, named pCAGGS-GM. All of our restriction enzyme digestion and cloning procedures were performed according to standard protocols.

Transfection

BHK-21 and 293T cells which grew to 90% confluence in 6-well cell culture plates were transfected with plasmid by Lipofectamine 2000 Reagent (11668-027; Invitrogen, Carlsbad, CA, USA) as previously described (Qi et al., 2008). Briefly, 2 μg of DNA was diluted in 50 μl of serum-free Opti-MEM medium, and Lipofectamine 2000 was diluted in 50 μl of serum-free Opti-MEM medium according to the manufacturer’s recommendation. We then added diluted DNA to each tube of diluted Lipofectamine 2000 Reagent (1:1 ratio) and incubated it at 25 °C for 5 min. We removed the cell culture medium and washed with Opti-MEM. Then, DNA-reagent complex was added to cells. At the same time, 500 μl Opti-MEM also added and further cultured at 37 °C. After 6 h, the supernatant was removed, and the cells were further incubated with fresh medium with 2% FBS for 48 h.

Indirect immunofluorescence assays (IFA)

The cells were washed with phosphate-buffered saline (PBS) and then, fixed with 4% paraformaldehyde at room temperature for 30 min. Cell monolayers were permeabilized with 0.2% Triton X-100 for 10 min. Then, incubated for 1 h in the presence of Mouse polyclonal antibodies against the NS1 protein of JEV (1:100) in PBS buffer at 37 °C, followed by 1 h incubation in PBS containing goat anti-mouse secondary antibodies conjugated to FITC at a dilution of 1:200. The fluorescence signals were visualized using a fluorescence microscope.

Western blotting

Western blot analysis was performed as previously described (Qi et al., 2015). Briefly, cell monolayers were incubated with RIPA lysis buffer (Beyotim, P0013B) containing 1 mM PMSF (ST506; Beyotim, Shanghai, China), then centrifuged and boiled with SDS-PAGE loading buffer. Equivalent Proteins sample were separated on SDS-PAGE gels and electro-transferred onto PVDF membranes (FFP30; Beyotim, Shanghai, China). After blocking, the membranes were incubated with primary antibodies, PRRSV GP5 protein monoclonal antibodies, and IRDye 800-conjugated goat anti-mouse as the secondary antibody. Finally, membranes were scanned using an Odyssey Imaging System (Li-Cor, Lincoln, NE, USA).

Immunization of mice

Sixty (eight-week-old) SPF female BALB/c mice were purchased from the Guangdong Medical Laboratory Animal Center (GDMLAC) and the number of production license is SCXK 2013-0002. BALB/c mice were assigned randomly into five groups of twelve mice each. Three groups of BALB/c mice were immunized 50 μl each hind leg with pJEV-REP-G-2A-M-IRES, pJEV-REP-G-2A-M, or pJEV-REP-IRES respectively (100 μg per mouse). Control groups were immunized using the same protocol with either pCAGGS-GM or PBS. Mice were immunized on days 0, 21, and 42. Mice blood samples were harvested every two weeks, and serum samples from each group were stored for virus antibody testing.

Enzyme linked immunosorbent assay (ELISA)

Mouse sera were obtained from blood samples. The PRRSV antibodies level was tested using the ELISA Kit (LSIVet Porcine PRRSV/US-Serum; Thermo Fisher, Waltham, MA, USA). Briefly, mouse sera were diluted 100-fold, and a 100 μl sample added to the 96 plate which are pre-packaged with PRRSV components. The wells were incubated for 30 min at 37 °C. Then, after washing with PBS, 100 μl HRP-Conjugate anti-mouse antibody reagent was added to every well, and incubated for 60 min at 37 °C. They were then washed again with PBS. A total of 50 ul Chromogen Solution A and Chromogen Solution B were added and evaded the light preservation for 15 min at 37 °C. Finally, the absorbance at 630 nm was read after adding the stop solution.

CCK-8 assay

Lymphocytes were isolated from different vaccination groups of mice two weeks after the third immunization and treatment as previously described (Cao et al., 2011). The WST-8 dye (Beyotime Institute Biotech, Jiangsu, China) was used to detect cell proliferation according to manufacture’s instruction. Briefly, lymphocytes isolated from different vaccination groups mice. The lymphocytes were adjusted to a working concentration of 1 × 106 cells/ml in RPMI-1640 and 100 μl lymphocytes were added into 96-well plate respectively. Then, 100 μl concentration of 200 TCID50/ml, XH-GD, PRRSV were added. As a control, 100 μl RPMI-1640 and 100 μl equal amount lymphocytes of PBS control group were added into another 96-well plate. Incubated the plates at 37 °C in a 5% CO2 incubator. Then, 10 μl WST-8 dye was added to each well and the 96-well plates were incubated at 37 °C for 3 h. The asorbance was determined at 450 nm using a microplate reader (Thermo Scientific, Waltham, MA, USA). Calculated the stimulation index (SI) on behalf of Proliferation of the T lymphocyte, which is the ratio of OD450 nm of stimulated wells to OD450 nm of unstimulated ones.

Plus or minus detection replicon RNA

The cells were harvested at the indicated times post transfection and total RNA was extracted using TaKaRa MiniBEST Universal RNA Extraction Kit (TaKaRa, Otsu, Shiga, Japan). The primer 2403F was used as an antisense primer and the primer 2609R as a sense reverse transcription, then 85 °C deactivation. The primers 2403F and 2609R (shown in Table 2) target a region within the E gene. After reverse transcription, the primer 2403F was used to selectively quantitate the sense strand and used primer 2609R to selectively quantitate the antisense strand by PCR.

Statistical analysis

For all the data, the mean value, the standard error of the mean (SEM), and statistical analyses were performed using both the unpaired t test and paired t test to determine the statistical significance (P < 0.05) between the two indicated test groups (GraphPad Prism 5.01 software).

Ethics statement and biosafety

Sixty (eight-week-old) SPF female BALB/c mice were purchased from the Guangdong Medical Laboratory Animal Center (GDMLAC); the number of production license is SCXK 2013-0002. All experiments were carried out in ABSL-3 facilities in compliance with the biosafety committee of South China Agriculture University (SCAU) protocols. All animal experiments were reviewed and approved by the Institutional Animal Care and Use Committee at SCAU and were carried out in accordance with the approved guidelines. All animal experiments were reviewed and approved by the Institutional Animal Care and Use Committee at SCAU and were carried out in accordance with the approved guidelines.

Results

Construction and characterization of JEV replicons

We constructed a DNA-based replicons vector, pJEV-REP, from the JEV SA-14-14-2 strain. pJEV-REP contains the JEV genome sequence with a large in frame deletion (nucleotides 165–2402) in the viral genome, corresponding to the middle portion of structure proteins, C-prM-E. The overall strategy for constructing the JEV replicon is outlined in Fig. 1A. The sequences of ribozyme HDVr and polyA had been added at the end of 3′UTR to insure that transcript RNA correctly cleaved within complete 3′UTR sequence. For convenience in inserting the necessary foreign genes, the SpeI and SalI restriction sites were introduced into the deletion position of the structure protein because those restriction enzymes have many isocaudomers (Sun et al., 2013).

To investigate the function of JEV replicons, two different versions of the replicons vectors were constructed, as shown in Figs. 1B and 1C. The replicons vector, pJEV-REP-GFP, was modified by inserting reporter gene, EGFP, into the SpeI and SalI restriction sites (Fig. 1B), while pJEV-REP-GFP-IRES was constructed by inserting the EGFP- IRES gene and introducing a termination codon and an initiation codon at the end of EGFP and the front of E25, respectively (Fig. 1C).

The efficiency and replication of JEV replicons

To investigate the replication of JEV replicons, pJEV-REP-GFP-IRES and pJEV-REP-GFP were transfected into 293T cells, and 48 h post transfection we applied indirect immune fluorescence analysis (IFA) to monitor viral protein NS1 synthesis. The results of IFA showed that IFA staining was readily detected (Fig. 2A). To facilitate replication monitoring and to explore the possibility of expressing foreign genes in the JEV replicons, the EGFP was used as a report gene and plasmids pJEV-REP-GFP-IRES and pJEV-REP-GFP was transfected into 293T cells and BHK-21cells, respectively. Following 48 h post transfection, more EGFP-expressing cells were detected in pJEV-REP-GFP-IRES than in pJEV-REP-GFP and the fluorescence intensity increased as time went by (Fig. 2B). In addition, the RT-PCR was performed to detect the replication of replicon RNA. The RT-PCR results showed increased levels of both plus- and minus-sense RNA, while no specific bands can be detected in the group without RT step (Fig. 3). Furthermore, when we diluted the cDNA at 10 times and 20 times, the specific bands amplified from plus-sense RNA were significantly brighter than those of minus-sense RNA. These results indicate that JEV replicons plasmids pJEV-REP-GFP-IRES and pJEV-REP-GFP replicated efficiently in transfected cells and an excess amount of plus-sense RNA was synthesized relative to the minus-sense RNA.

Figure 2 The efficiency and replication of JEV replicons.

(A) 293T cells transfected with replicon plasmids pJEV-REP-GFP-IRES or pJEV-REP-GFP were subjected to IFA at the indicated time points post transfection. JEV-NS1 polyclonal antibody and FITC-conjugated goat anti-mouse IgG antibody were used as primary and secondary antibodies for IFA, respectively. (B) BHK-21 cells and 293T cells were transfected with pJEV-REP-GFP-IRES or pJEV-REP-GFP and monitored the EGFP signals by fluorescence microscope at 48 h post transfection.

Figure 3 Plus or minus detection of replicon RNA.

(A) BHK-21 cells were transfected with replicon plasmids pJEV-REP-GFP-IRES or pJEV-REP-GFP. Total RNA isolated at 48 h post transfection was subjected to RT-PCR analysis with JEV-specific primers. RT reaction was performed under standard conditions with avian myeloblastosis virus (AMV) reverse transcriptase (Takara, Otsu, Shiga, Japan), Primer 2403F and 2609R targeting a region within the E gene, was used as a sense or an antisense primer respectively in RT step to selectively quantitate either minus- or plus-sense replicon RNA. (B) The cDNA samples obtained from B were diluted with 10 times or 20 times and quantitate either minus- or plus-sense replicon RNA by PCR.

Generation of recombinant JEV replicons expressing GP5/M protein

The JEV replicon constructed above was used for the generation of recombinant vectors expressing the GP5/M gene of PRRSV. The nucleotide sequence of the HP-PRRSV strain, XH-GD, ORF5 encoding GP5 protein (almost 22KD) and ORF6 encoding M protein (almost 19KD) were amplified from the HP-PRRSV strain XH-GD genome. A DNA fragment encoding 2A protease of the foot-and-mouth disease virus (FMDV-2A) was inserted between proteins GP5 and M. Fragments G-2A-M and G-2A-M-IRES were cloned into the pJEV-REP vector and named pJEV-REP-G-2A-M and pJEV-REP-G-2A-M-IRES, respectively (Fig. 4A). The G-2A-M gene was also cloned into a eukaryotic expression vector, pCAGGS, named pCAGGS-GM to be used as control. To verify the GP5/M protein expression, pJEV-REP-G-2A-M-IRES or pJEV-REP-G-2A-M was transfected into 293T cells and transfected cells were harvested and analyzed by IFA and western blot. IFA results showed that NS1 was expressed (Fig. 4B), and western blot analysis showed that in the pJEV-REP-G-2A-M, pJEV-REP-G-2A-M-IRES and pCAGGS-GM, the fusion protein were expressed in 293T cells successfully and in negative control the fusion protein were not detected (Fig. 4C).

Figure 4 Expression of PRRSV GP5/M protein in JEV replicons.

(A) Schematic drawing of pJEV-REP-G-2A-M and pJEV-REP-G-2A-M-IRES. (B) 293T cells transfected with replicon plasmids pJEV-REP-G-2A-M and pJEV-REP-G-2A-M-IRES were subjected to IFA at the indicated time points post transfection. JEV-NS1 polyclonal antibody and FITC-conjugated goat anti-mouse IgG antibody were used as primary and secondary antibodies for IFA. (C) Western Blot was used to analyze the GP5 and M protein expression, PRRSV GP5 protein monoclonal antibodies were used as primary antibodies and IRDye 800-conjugated goat anti-mouse as the secondary antibody. 293T cells transfected with pCAGGS-GM was used as a positive control while 293T cells were used as a negative control. In the pJEV-REP-G-2A-M, pJEV-REP-G-2A-M-IRES and pCAGGS-GM, the fusion protein were expressed in 293T cells successfully, and in negative control the fusion protein were not detected.

Cellular immune response induced by recombinant JEV replicons expressing GP5/M protein

To characterize the cell-mediated immune response induced by the JEV replicons vaccine, mice were killed eight weeks after the primary immunization, and the splenocyte proliferation-based WST-8 assay was performed. WTS-8 assay indicated that the lymphocyte proliferation of mice immunized with pJEV-REP-G-2A-M-IRES or pJEV-REP-G-2A-M was higher than those immunized with pCAGGS-GM (DNA vaccine), but not significantly so (Fig. 5). Lymphocyte proliferation of the three immunized groups, pJEV-REP-G-2A-M-IRES, pJEV-REP-G-2A-M and pCAGGS-GM, were significantly high than the control groups pJEV-REP-IRES or PBS. These results indicate that immunization with the JEV DNA-based replicon vaccine induces lymphocyte proliferation in mice.

Figure 5 Lymphocyte proliferation elicited by recombinant JEV replicons expressing GP5/M protein in mice.

Eight-week-old, SPF BALB/c mice were immunized with pJEV-REP-G-2A-M-IRES, pJEV-REP-G-2A-M, pCAGGS-GM, pJEV-REP-IRES, and PBS respectively. Lymphocytes were separated from the spleens of mice two weeks after the third immunization. Then, 100 μl lymphocytes were cultured with 100 μl PRRSV. 100 μl RPMI-1640 and 100 μl lymphocytes of PBS control group were added into another 96-well plate as a control. After 84 h, CCK-8 assay was performed to detect the lymphocyte proliferation. Stimulation Index (SI) = OD450 nm (PRRSV)/OD450 nm(Control). Statistical were compared the pCAGGS-GM, pJEV-REP-G-2A-M-IRES and pJEV-REP-G-2A-M groups with the pJEV-REP-IRES group, respectively. Data were shown as Mean + SEM. Statistical comparisons were made between each JEV replicon vaccine group and the control group pJEV-REP-IRES or PBS (n = 3; *, p < 0.05, t test).

PRRSV-specific antibodies elicited by recombinant JEV replicons expressing GP5/M protein in mice

Sera were harvested from mice at 0, 2, 4, 6, 8, and 10 weeks after the first immunization and PRSSV-specific antibodies were analyzed using ELISA. As shown in (Fig. 6), mice immunized with pJEV-REP-G-2A-M-IRES or pJEV-REP-G-2A-M could produce higher PRRSV specific antibodies than those immunized with pCAGGS-GM two weeks after the first immunization, and the specific antibodies of the three groups immunized with pJEV-REP-G-2A-M-IRES or pJEV-REP-G-2A-M or pCAGGS-GM peaked two weeks after the third immunization. Anti-PRRSV specific antibodies titers were significantly (P < 0.05) higher in immune groups pJEV-REP-G-2A-M-IRES and pJEV-REP-G-2A-M than in the control groups pJEV-REP-IRES or PBS six weeks after first vaccination, but the pCAGGS-GM group was non-significant (P > 0.05). However, Anti-PRRSV specific antibodies titers were significantly (P < 0.05) higher in all three immune groups, pJEV-REP-G-2A-M-IRES, pJEV-REP-G-2A-M, and pCAGGS-GM, than in the control groups pJEV-REP-IRES or PBS at eight weeks after the first immunization. Although the antibody level declined, anti-PRRSV specific antibodies titers were also significantly (P < 0.05) higher in the immune groups pJEV-REP-G-2A-M-IRES and pJEV-REP-G-2A-M than in the control groups pJEV-REP-IRES or PBS at 10 weeks after the first immunization.

Figure 6 Specific antibodies elicited by recombinant JEV replicons expressing GP5/M protein in mice.

Eight-week-old, SPF BALB/c mice were immunized with pJEV-REP-G-2A-M-IRES, pJEV-REP-G-2A-M, pCAGGS-GM, pJEV-REP-IRES, and PBS respectively. Mice sera were collected at indicated time points after immunization and used to detect the GP5 specific antibodies by ELISA. Data were shown as Mean + SEM. Statistical comparisons were made between each JEV replicon vaccine group and the control group pJEV-REP-IRES or PBS (n = 3; *, p < 0.05, t test).

Discussion

In the swine industry today, HP-PRRS is considered to be one of the most challenging diseases. Vaccination has been an effective method of controlling PRRS ever since it was reported (Pileri & Mateu, 2016). However, both MLVs and inactivated virus vaccines have inherent drawbacks. Furthermore, like other (+) RNA viruses, PRRSV is easy to mutate and recombine. In 2013–2014, a new HP-PRRSV strain emerged in China with a very different genetic background than the classic Chinese HP-PRRSV strains. It is a NADC30-like PRRSV strain recently introduced from North America that has undergone genetic exchange with the classic HP-PRRSV strains (Zhao et al., 2015), and the occurrence of attenuated strains reverting to high virulence strains has been reported (Jiang et al., 2015). Thus, it is necessary to develop effective, safe, and quickly obtained vaccines to protect against PRRSV, especially when new PRRS strains emerge.

The GP5 and M proteins are two kinds of structure proteins of the PRRSV, and they were associated in hetero dimeric complexes on the surface of PRRSV (Mardassi, Massie & Dea, 1996). The GP5 protein and M protein have been shown to induce antibodies and high production of IFN- β (Binjawadagi et al., 2016). Here, we describe the construction of JEV replicons and use JEV replicon vectors expressing the PRRSV GP5 and M proteins as a bivalent seedlings vaccine. We generate two kinds of PRRS vaccine, named pJEV-REP-G-2A-M-IRES and pJEV-REP-G-2A-M. In order to release the GP5 and M proteins from the JEV replicon polyprotein, the foot-and-mouse disease virus 2A autoprotease sequence was inserted between the GP5 and M genes. IFA with JEV-NS1 polyclonal antibody show that the JEV replicon vector can replicate effectively (Fig. 4B). Western blot analysis showed that GP5/M proteins were also expressed successfully in vitro (Fig. 4C). Splenocyte proliferation was an important point in detecting a cell-mediated immune response. The result has demonstrated the ability of JEV replicon vaccine to induce splenocyte proliferation following the final immunization. The ability to induce PRRSV specific immune responses was detected by ELISA. The research showed that the intramuscular immunization of mice with the JEV vaccine induced special anti-PRRSV antibodies after the first inoculation. With subsequent immunization boosting, the level of specific anti-PRRSV antibody increased and finally peaked two weeks after the third immunization. The result showed that the antibody levels of immune groups pJEV-REP-G-2A-M-IRES and pJEV-REP-G-2A-M were higher than immune group pCAGGS-GM all the time, and the antibodies of immune group pCAGGS-GM decreased faster than the JEV replicon vaccine group. Furthermore, in the pJEV-REP-IRES vaccinated group, the level of special anti-PRRSV antibodies were the same as in the PBS group. Meanwhile, the ELISA method is very accurate and the immune response is specifically directed against PRRS proteins. We can infer that the PRRS proteins GP5 and M were responsible for the observed immune response rather than the backbone of the JEV replicon. Of course, the ELISA method is very accurate. However, an additional control with pJEV-REP-IRES plasmid expressing an irrelevant protein is worth taking into account to rule out the possibility that any protein different from GP5 or M induces antibodies that cross-react with PRRSV antigens. All the data showed that the JEV replicon vaccine induced an effective antibody response against PRRSV. However, the protection capability of the JEV replicon vaccines against PRRS needs more assay in pigs, although previous report showed that pigs immunized with CSF-JE VRP replicon vaccine displayed strong antibody responses and protection against CSFV and JEV challenge infections (Yang et al., 2012). In the aspect of preventing JEV infection, it was reported that the JEV replicon vaccine could confer protection to itself (Huang et al., 2015). Therefore, we focus on the immune protection to PRRS rather than JEV. Of course, this genetic engineering vaccine can be further optimized to enhance the immune protective effect; for example, to screen appropriate immune adjuvant for JEV replicon vaccine or inoculation of animals with suitable methods. At the same time, the vaccine with JEV replicons expressing GP5/M proteins induced a systemic immune responses to PRRS; it could be a good heterologous prime-boost HP-PRRS vaccination regimen. The JEV replicon vaccine can be used for the first immunization and the commercial HP-PRRS vaccines can be used to improve anti-PRRSV immunity. Because it is laborious to attenuate PRRSV and it takes a long time to develop a new modified live-attenuated vaccine when a new variation of PRRS emerges, this genetically engineered vaccine is a good emergency supplement strategy.

Flaviviruses replicons, characterized by their high efficiency in expressing heterologous genes without producing infectious progeny virus, are useful tools for understanding the replication of viruses and exploring antiviral screening, and can also be applied as a potential expression system to be an antivirus vaccine candidate (Cao et al., 2011; Suzuki et al., 2014). Replicons vaccines have several advantages over inactivated vaccines and many subunit vaccines. They can be prepared more quickly than inactivated virus vaccines, and replicons vaccines can enhance cross-protection through the fusion of expressing antigenic peptides of different strains (Sun et al., 2016). Compared with the baculovirus expression system, replicon vaccines are more convenient to operate since they do not need to express antigens by cell culture and be purified. Otherwise, the baculovirus expression system is liable to fail in vivo because of the complement system of the host (Tani et al., 2003). Therefore, we constructed JEV DNA-based replicons through deleting the C, prM, E encoding region which replicates autonomously but fails to generate infectious virus progeny in non-complementing cells; therefore, they can be used as an expression system for foreign proteins. In this study, the nucleotides from 165–2402, corresponding to most of the structural proteins C, prM, E, were deleted in JEV replicons. The N-terminal 23 amino acids of the C protein, have been reported as performing the essential role of the cis-acting element and in regulating minus sense RNA synthesis (also known as containing the cyclization sequence) (Khromykh et al., 2001). Compared with other JEV replicons, we retained less of the nucleic acid sequence of the C protein, and the IFA and RT-PCR results showed that our constructed JEV replicon can self-replicate effectively (Fig. 2). We also retained the C-terminal 25 amino acids of the E protein to preserve the correct processing and translocation of NS1 and the remaining nonstructural polyprotein in the correct topology across the membrane of the endoplasmic reticulum (Ng et al., 2007). At the same time, we constructed JEV DNA-based replicons with a cytomegalovirus (CMV) promoter. Compared with RNA-based replicons, DNA-based replicons operate more conveniently and are more stable in the form of plasmids. Furthermore, host cells are more likely to intake DNA-based replicons (Cao et al., 2011; Varnavski, Young & Khromykh, 2000).

Conclusions

In conclusion, we describe the construction of JEV replicons with deletion in the C, prM, E encoding region, which can be used as an efficient expression system for foreign proteins. In addition, this JEV replicon used to express GP5/M proteins which showed better immunogenicity compared with tradition DNA vectors expressing GP5/M proteins. These results indicate that our JEV replicons are a useful molecular platform for expressing foreign proteins capable of inducing a protective immune response and could serve as a promising strategy in developing a potential bivalent seedlings vaccine candidate.

Supplemental Information

Supplemental Information 1 Raw data of WST-8 assay

Lymphocyte proliferation elicited by recombinant JEV replicons expressing GP5/M protein in mice. Eight-week-old, SPF BALB/c mice were immunized with pJEV-REP-G-2A-M-IRES, pJEV-REP-G-2A-M, pCAGGS-GM, pJEV-REP-IRES, and PBS respectively. Lymphocytes were separated from the spleens of mice two weeks after the third immunization. Then, lymphocytes were cultured with PRRSV. As a control, RPMI-1640 and lymphocytes of PBS control group were added into another 96-well plate. After 84h, CCK-8 assay was performed to detect the lymphocyte proliferation. Stimulation Index (SI) =OD450 nm (PRRSV)/ OD450 nm (Control).

Click here for additional data file.

Supplemental Information 2 Raw data of ELISA to detect specific antibodies

Specific antibodies elicited by recombinant JEV replicons expressing GP5/M protein in mice. Eight-week-old, SPF BALB/c mice were immunized with pJEV-REP-G-2A-M-IRES, pJEV-REP-G-2A-M, pCAGGS-GM, pJEV-REP-IRES, and PBS respectively. Mice sera were collected at indicated time points after immunization and used to detect the GP5 and M specific antibodies by ELISA.

Click here for additional data file.

We would like to thank the National and Regional Joint Engineering Laboratory for Medicament of Zoonoses Prevention and Control providing HP-PRRSV strain. Also, we would like to thank Harbin Veterinary Research Institute for providing plasmids. Moreover, we would like to thank Shanghai Veterinary Research Institute for providing antibodies. All authors reviewed and revised the first and final drafts of this manuscript.

Additional Information and Declarations

Competing Interests

Author Contributions

Animal Ethics

Data Availability

Wenbao Qi is an Academic Editor for PeerJ.

Pingsheng Hu, Xiaoming Chen, Lihong Huang and Shukai Liu performed the experiments, analyzed the data, contributed reagents/materials/analysis tools, wrote the paper, prepared figures and/or tables, reviewed drafts of the paper.

Fuyu Zang performed the experiments, contributed reagents/materials/analysis tools, prepared figures and/or tables.

Jinchao Xing, Youyue Zhang and Jiaqi Liang performed the experiments, contributed reagents/materials/analysis tools.

Guihong Zhang analyzed the data, contributed reagents/materials/analysis tools, reviewed drafts of the paper.

Ming Liao and Wenbao Qi conceived and designed the experiments, analyzed the data, wrote the paper, prepared figures and/or tables, reviewed drafts of the paper.

The following information was supplied relating to ethical approvals (i.e., approving body and any reference numbers):

South China Agricultural University provided experimental animal welfare ethical approval for this research.

The following information was supplied regarding data availability:

The raw data has been provided as Supplemental Files.

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
