# Peer review of "A highly pathogenic porcine reproductive and respiratory syndrome virus candidate vaccine based on Japanese encephalitis virus replicon system"

_PeerJ, doi:10.7717/peerj.3514_

## Round 0.1 · original submission · Minor Revisions

· Academic Editor

Minor Revisions

Should you respond and modify, when neded, to every one of the reviewers comments, I will be happy to receive your revised version.

·

Basic reporting

Although the English language allows the reader to understand the main ideas and explanations throughout the text, which means that is clear and unambiguous, there are some writting issues that could create confusion in relevant parts of the text. Some examples of these writting issues are:
1. Line 23: JEV stands for Japanese encephalitis virus, but the first is used, it shouldn't be abbreviated.
2. Line 27: "structural (C-prM-E) of the genome" could be substituted by "structural genes (C-prM-E)"
3. Line 43-44: "The GP5 protein of PRRSV (encoded by ORF5), which is one of the
major antigens expressed on virion surfaces and combines with matrix (M) protein, are thought to be the most important targets of protective antibodies" could be substituted by
"The GP5 protein of PRRSV (encoded by ORF5) is one of the major antigens expressed on virion surfaces and along with matrix (M) protein, are thought to be the most important targets of protective antibodies"
4. Line 63: "which are not infectious" could be substituted by "without being infectious"
5. Line 65: "it can elicit" could be substituted by "they can elicit"
6. Line 76: "base" could be substituted by "based"
7. Line 135-136: "Removed the cell culture medium and washed with Opti-MEM." can be
substitutes by "We removed the cell culture medium and washed with Opti-MEM."
8. Line 164: "test" could be substituted by "tested"
9. Line 185: "extract" could be substituted by "extracted"
10. Line 215: "termination code and an initiation code" could be substituted by "termination codon and an initiation codon"
A minor issue is that the manuscript is written sometimes in first person plural and others in indeterminate. it is advisable to use either one throughout the whole document.
I strongly suggest that a native English speaking person review the document.

Experimental design

There are some concerns that, once addressed, will contribute to increase the scientific value of the findings:
1. In the experiment that demonstrates GP5 and M expression in the transfected cell line by Western Blot, authors may perform, if possible, a Western Blot for detecting a housekeeping protein as protein loading control. This will demonstrate that same amount of protein was loaded in each lane. Otherwise, I suggest you to mention in the manuscript that care was taken in loading the same amount of protein per lane.
2. In the CCK-8 proliferation experiment, you mentioned the use of volumes for both lymphocytes and PRRSV virus. It is simple but so important to report the amount of lymphocytes used in the experiment. The same applies for PRRSV particles numbers. Also, it is necessary to mention which PRRSV strain was used.
3. It is neccesary to specify the anti-PRRSV antibodies that were used to detect GP5 and M proteins in Panel C of Figure 4.
4. In the specific PRRSV antibody response in mice, an additional control with pJEV-REP-IRES plasmid expressing an irrelevant protein would have been better than using pJEV-REP-IRES plasmid in order to rule out the possibility that any protein different from GP5 or M induces antibodies that cross-react with PRRSV antigens. Although the possibility that this happens is low, it is worth it to demonstrate it. Otherwise, you must explain this on the manuscript.

Validity of the findings

1. Results derived from panel C of Figure 4 describing PRRSV GP5 and M protein expression are poorly explained. What is the expected size for each protein? GP5 protein is normally 20 kDa and M protein is 17 KDa. I suggest you a longer description of this figure on the Results section.
2. CCK-8 proliferation experiment: normalization based on lymphocyte numbers should be included. This information is missing.
3. Specific PRRSV antibody response in mice: An additional control with pJEV-REP-IRES plasmid expressing an irrelevant protein would have been better than using pJEV-REP-IRES plasmid in order to rule out the possibility that any protein different from GP5 or M induces antibodies that cross-react with PRRSV antigens. Although the possibility that this happens is low, it is worth it to demonstrate it. Otherwise, authors should explain this on the manuscript. In the same experiment, authors mention that LSIVet Porcine PRRSV/US-serum ELISA plate is pre-packaged with GP5 antibody. It looks to me that this ELISA may be pre-packaged with PRRSV components, as it is intended for detection of antibodies against the virus. In this case, manuscript should be modified at some parts according to this observation.

Comments for the author

I really think that the manuscript is of high scientific value. The use of a flavivirus backbone for the expression of GP5 and M proteins from PRRSV is perfectly justified and opens the possibility of using this expression platform for speedy upgrades of a PRRSV vaccine, which, in my opinion, will need to be done every once in a while due to the virus nature. If the above-mentioned concerns are properly addressed, the manuscript is suitable for publication in this Journal. I strongly encourage you to address these issues, most of them are minor ones.

Reviewer 2 ·

Basic reporting

Authors describe the construction of novel vaccine based on Japanese encephalitis virus. this vaccine express proteins derived from PRRS virus. The report is well documented and clear. Literature cited is sufficient and figures are well prepared. Minor modifications are necessary, in this context remplace the word "Special" antibodies for "specĂ­fic" antibodies. i.e. line 266, 269, 272, 303.

Major changes.
line 304, Author declare that JEV construction were better in inducing immune response than DNA vaccine (PCAGGS-GM), and antibodies in the latter decreased faster in comparison with JEV immunized mice. Why this declarations is Obvious? It is well documented that it depends of the construction, animal model among others. Maybe the word, "obviously" should be deleted or argument the response.

Line 309. Why the author infer that immune response is directed by PPRS proteins rather to JEV backbone? I think, because the ELISA method selected, the immune response is specifically directed against PRRS proteins. However, author should demonstrate this issue.

line 310-311. Author do not present data on viral neutralization or challenge, for his reason, I suggest author delete the word "protection".

313. Author should explain the risk of this vectors, in specific, the vaccine is suggested to be used in pig, in pigs, specific immunity agains JEV has been demonstrated? If true, this kind of vector could be suitable for vaccination? Previos reports indicated that preexisting immunity against against flavivirus diminish the efficacy of viral vaccines.

line 318. Why author argument that this novel vaccine is not intended as substitute? Maybe author need to rewrite this part or argument the major disadvantages of the vector.

line 352. Again, the word "protective" must be replaced. i.e. better immunogenicity.

Experimental design

An additional experiment should be performed, the demonstration that sera obtained from vaccinated mice is able to recognize the proteins of PRRS in WB. Although not necessary because the ELISA method is highly specific for the gp5, this new experiment could confirm the reactivity of these generated sera.

Validity of the findings

no comments

Comments for the author

The report is suitable for publication after clarify the observations.

---

## Round 0.2 · accepted · Accept

· Academic Editor

Accept

I am glad to see that you attended all suggestions provided by referees. I only would like to recommend another revision of grammar by a native speaker or professional service.

·

Basic reporting

The suggested changes regarding mistyping and grammar mistakes now have been made. There were no suggestions related to literature references, article structure.

Experimental design

As in previous section, the authors have put attention to the corresponding suggestions and have included explanations in the manuscript.

Validity of the findings

Now the manuscript is covering all issues related to the validity of their findings. I am totally satisfied with this revised version.

Comments for the author

There are no comments for the authors.

Reviewer 2 ·

Basic reporting

No comments

Experimental design

No comments

Validity of the findings

No comments

Comments for the author

Author have answer all the issues made in the past revision, in the same way have rewritten some paragraphs that complements the manuscript. I feel the paper is suitable for publication.